# Classical criticality via quantum annealing

Pratik Sathe [1,2,7] ✉, Andrew D. King [3], Susan M. Mniszewski [4], Carleton Coffrin[5], Cristiano Nisoli [1] ✉ & Francesco Caravelli[1,6] ✉

Quantum annealing provides a powerful platform for simulating magnetic materials and realizing statistical physics models, presenting a compelling alternative to classical Monte Carlo methods. We demonstrate that quantum annealers can accurately reproduce phase diagrams and simulate critical phenomena without suffering from the critical slowing down that often affects classical algorithms. To illustrate this, we study the piled-up dominoes model, which interpolates between the ferromagnetic 2D Ising model and Villain's fully frustrated "odd model". We map out its phase diagram and employ finite-size scaling and Binder cumulants on a quantum annealer to study critical exponents for thermal phase transitions. Our method achieves systematic temperature control by tuning the energy scale of the Hamiltonian, eliminating the need to adjust the physical temperature of the quantum hardware. This work demonstrates how, through fine-tuning and calibration, a quantum annealer can be employed to apply sophisticated finite-size scaling techniques from statistical mechanics. Our results establish quantum annealers as robust statistical physics simulators, offering a novel pathway for studying phase transitions and critical behavior.

Quantum simulation–the realization of models of collective, interacting qubits within a quantum machine–has been a central motivation in the development of quantum computing and quantum information science[1–3]. Numerous digital as well as analog quantum simulation approaches have been implemented successfully over the last few decades[4]. Among them, quantum annealing-an analog method grounded in the adiabatic theorem-was originally introduced to solve classical combinatorial optimization problems[5–8]. However, when repurposed for analog simulation, superconducting qubit-based quantum annealing (QA) has shown promise in simulating a range of quantum condensed matter systems[9–17]—in some cases, potentially outperforming the best-known classical algorithms[18,19] (see also Refs. 20, 21 for critical perspectives).

While quantum simulation applications of QA have received significant attention (particularly in the so-called coherent regime in which coupling to the environment is negligible[13,15,17]), we propose and demonstrate that QA can be used to simulate thermodynamic phase transitions in classical statistical physics models when operated in the incoherent regime. The "incoherent regime" usually corresponds to long annealing times, of the order of tens to hundreds of microseconds, thus leveraging a perceived drawback as an advantage. In this regime, quantum annealers have been shown to sample from the equilibrium (i.e., Gibbs or Boltzmann) distributions corresponding to the programmed classical Ising models, due to a combination of non-adiabaticity and environmental interactions[22–25].

A central goal in the study of statistical physics models is the identification of critical points and their associated critical exponents. In the absence of exact solutions–which is often the case–Markov-Chain Monte Carlo (MCMC) methods are a standard numerical approach[26–28]. These algorithms aim to generate and analyze samples from equilibrium distributions across different temperatures and parameter regimes. Given its ability to perform a similar sampling task, QA offers a natural alternative for simulating classical systems[29].

[1]Theoretical Division, Quantum & Condensed Matter Physics, Los Alamos National Laboratory, Los Alamos, NM, USA. [2]Information Science & Technology Institute, Los Alamos National Laboratory, Los Alamos, NM, USA. [3]D-Wave Quantum Inc., Burnaby, BC, Canada. [4]Computing and Artificial Intelligence (CAI) Division (CAI-3 Information Sciences), Los Alamos National Laboratory, Los Alamos, NM, USA. [5]Advanced Network Science Initiative, Los Alamos National Laboratory, Los Alamos, NM, USA. [6]Dipartimento di Fisica dell'Università di Pisa, Pisa, Italy. [7]Present address: D-Wave Quantum Inc., Burnaby, BC, Canada. ✉e-mail: psathe@dwavesys.com; cristiano@lanl.gov; francesco.caravelli@proton.me

QA has previously been employed to study various aspects of classical statistical physics models, particularly those involving geometric frustration. Examples include reproducing the ground-state ($T = 0$) phase diagram of the (classical) Shastry-Sutherland model[30]; measuring order parameters as functions of tuning parameters at an unknown but fixed temperature[31,32]; investigating artificial spin-ice dynamics[33–35]; and extracting critical exponents in classical 3D spin-glass models under parameter variation at fixed temperature[9]. In some of these studies, Gibbs sampling is not explicitly invoked, and the focus is either on zero-temperature behavior or on sampling with model parameters varied at a fixed, device-dictated temperature.

Quantum annealing implementations often deviate from ideal Boltzmann sampling[36], exhibiting issues such as biased sampling within degenerate ground states[37–39], the influence of noise[40], and quantum effects[41]. Although high-quality Gibbs sampling has been demonstrated under specific conditions[42] and can be improved through specialized techniques[43,44], the identification of the precise conditions when that occurs– and achieving fine, systematic control over the sampling temperature– remains computationally expensive and generally infeasible for the large system sizes typically studied in statistical physics. Hence, despite its promise, QA has yet to be used to reconstruct full thermodynamic phase diagrams with thermal phase transitions without privileged access to the device's physical temperature.

In contrast, we show that the sampling temperature can be systematically controlled by tuning the Hamiltonian's energy scale. Despite known challenges in achieving high-quality Boltzmann sampling with QA, we demonstrate that QA can still capture thermodynamic phase transitions in classical statistical models and serve as a viable alternative to traditional Monte Carlo methods. To this end, we present a methodology for reconstructing phase diagrams and critical behavior in classical systems and apply this methodology to the Piled-Up Dominoes (PUD) model[45], which is a classically solvable spin system with tunable frustration. In addition to its rich phenomenology, this model played a key role in the theoretical development of the order-by-disorder phenomenon by inspiring Villain et al.'s domino model[46]. Moreover, its Ising form, regular lattice structure, and exact analytical solution make it an ideal testbed for benchmarking quantum annealers. Geometrically frustrated systems are generally more challenging to simulate numerically[26], and the tunable degree of frustration in the PUD model controls this sampling difficulty. The competing and mutually incompatible interactions that define geometric frustration can give rise to exotic phases such as spin ice and spin liquid phases[47]. The PUD model, therefore, serves as both a meaningful and demanding benchmark for assessing the effectiveness of our approach.

## Results

### The piled up Dominoes model

The PUD model is a family of Hamiltonians interpolating between Villain's fully-frustrated "odd model" (which does not have a phase transition)[48], and the ferromagnetic 2D Ising model (which does have a phase transition)[49], the interpolation being controlled by a parameter $s$:

$$H(s) = (1 - s)H_{\text{2D−Ising}} + sH_{\text{Villain}} \tag{1a}$$

$$= -\sum_{x,y} \sigma_{x,y}\sigma_{x,y+1} - \sum_{x,y} \sigma_{2x,y}\sigma_{2x,y+1} - (1 - 2s)\sum_{x,y} \sigma_{2x+1,y}\sigma_{2x+1,y+1}, \tag{1b}$$

where $\sigma_{x,y} = \pm 1$ denotes a (classical) Ising spin located at position $(x, y)$ on a square grid.

When visualized over a 2D grid (see Fig. 1a), all the horizontal links and the links on alternate vertical columns are ferromagnetic with a coupling strength of 1, with all the remaining vertical columns having coupling values equal to $1-2s$. We note that the sign convention for the Hamiltonian in terms of the exchange energy $J$ in Eq. 1, while standard in condensed matter physics, is the opposite of the one used in D-Wave's QA convention. The model, therefore, has tunable frustration. It has been solved exactly using transfer matrix methods[45], thus providing a good testbed to probe the potential of QA in the study of criticality. Specifically, expressions for the partition function with zero longitudinal field and the boundaries separating the three phases have been analytically derived. (The critical exponents have been analytically computed only at $s = 0$.) More recently it was also solved via a dimer mapping in Refs. 50 (wherein it was referred to as the Onsager-Villain model). Its rich phase diagram includes ferromagnetic, paramagnetic, and antiferromagnetic phases, separated by critical lines in the $s$-$T$ (where $T$ is temperature) plane (Fig. 1b).

To control the sampling temperature, we consider a simple model of D-Wave's QA devices in which the user inputs a Hamiltonian $H_{\text{input}}$, and the device samples from the corresponding Boltzmann distribution at a fixed, but unknown, device-dependent temperature $T_{\text{sampler}}$. Since $T_{\text{sampler}}$ typically depends on the anneal time and schedule, we fix these to be $100\mu s$ and standard forward anneals throughout our study. That is, the probability of generating a spin sample $\{\sigma_{x,y}\}$ is given by

$$p(\{\sigma_{x,y}\}) = \frac{1}{\mathcal{Z}}e^{-\beta_{\text{sampler}}H_{\text{input}}(\{\sigma_{x,y}\})}, \tag{2}$$

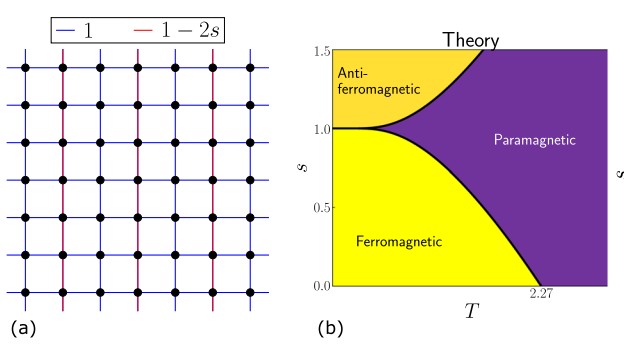

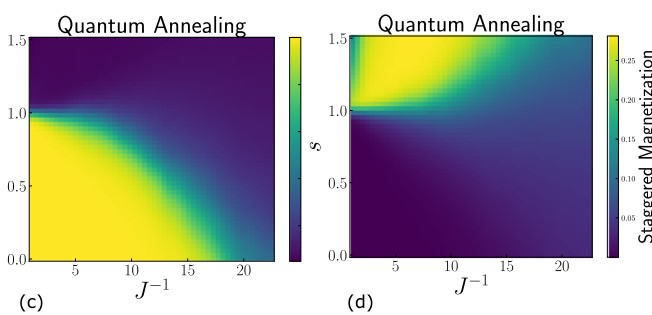

**Fig. 1 | The PUD model and its solution using theory and quantum annealing. a** The PUD model is an Ising model with a tunable degree of frustration. Defined on a square lattice, it has two types of nearest-neighbor couplings $J_{ij}$, shown here in blue and red, with the Hamiltonian defined as $H = -\sum_{i<j}J_{ij}s_is_j$. **b** The phase diagram in the $s$-$T$ plane obtained from the exact solution. Data corresponding to quantum annealing samples are shown in (**c**) and (**d**). Specifically, the ferromagnetic order parameter (sub-figure **c**) and the antiferromagnetic (or staggered) order parameter (sub-figure **d**) corresponding to samples generated using D-Wave's `Advantage2_prototype2.6` device for a toroidal system of size 12 × 12 are in qualitative agreement with the exact solution.

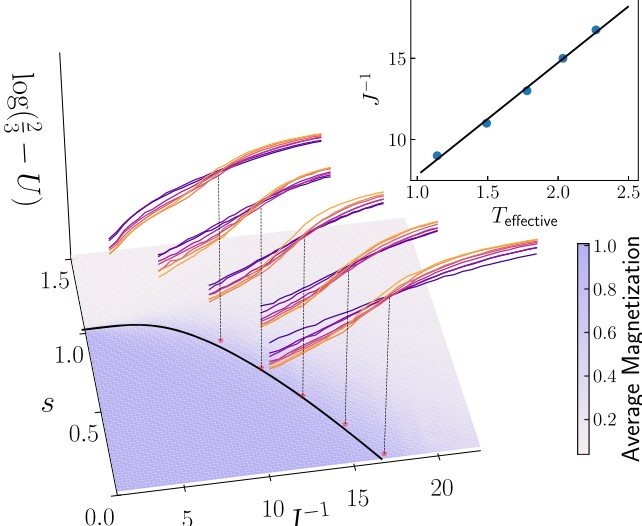

**Fig. 2 | Critical points extracted from Binder cumulant crossings overlaid on magnetization.** The critical inverse energy scale $J_c^{-1}$ is located for 5 values of $s$ using the intersection of Binder cumulant curves for various system sizes (indicated by different colors unrelated to the color-bar). Using the exact solution $T_c(s)$ and $J_c^{-1}(s)$ extracted from QA samples, we plot $J_c^{-1}$ as a function of the effective sampling temperature $T$. We find that $J_c^{-1}$ is linearly proportional to $T$, consistent with the initial assumption Eq. (3) (see inset). In the main figure, we show the ferromagnetic order parameter (i.e., $\langle |m| \rangle$) for a toroidal system of size $12 \times 12$, as a function of $s$ and $J^{-1}$, along with the $J_c^{-1}$ extracted from five different values of $s$. $J_c^{-1}(s)$ is inferred from the linear fit and the analytical expression for the critical line, and clearly separates the ferromagnetic phase from the paramagnetic phase.

where $\mathcal{Z}$ is the normalizing factor (also known as the partition function), and $\beta_{\text{sampler}} = 1/T_{\text{sampler}}$. To generate samples for an $H$ [such as the PUD model from Eq. (1b) at some value of $s$ of interest], we set $H_{\text{input}} = JH$ for some value of "energy scale" $J$. It follows from Eq. (2) that the generated samples correspond to the Hamiltonian $H$, but at inverse temperature $\beta_{\text{effective}} = J\beta_{\text{sampler}}$, or equivalently,

$$T_{\text{effective}} \propto J^{-1}. \tag{3}$$

Thus, increasing the inverse energy scale of the input Hamiltonian proportionally increases the sampling temperature of the device. For convenience, we will henceforth drop the subscript 'effective' while referring to the effective temperature and its inverse.

We note that the dependence of the effective temperature on the energy scale and annealing time has been studied previously in Ref. 42. The inverse energy scale was used as a proxy for the effective temperature in Ref. 35. However, prior studies either focused on small system sizes or assumed the validity of Eq. (3) without thorough verification. In this work, using the exact solution of PUD and the finite-size scaling method, we demonstrate the validity of Eq. (3) across a range of system sizes (see Fig. 2; details provided below).

The exact solution of the PUD model[45,50] yields an $s$-$T$ phase diagram (see Fig. 1b) that exhibits three distinct phases— ferromagnetic, antiferromagnetic and paramagnetic— separated by critical lines that emanate from the point ($s = 1$, $T = 0$).

## Phase Diagram

To reproduce this phase diagram using quantum annealers, we implement sampling experiments over a grid of values of $s$ (by programmatically controlling the relative coupler values) and over a grid of $T$ values [by tuning the energy scale of the Hamiltonian, as described in Eq. (3)].

Using D-Wave's `Advantage2_prototype2.6` quantum annealer, we successfully observe all three phases. (See the Methods section for details of the procedure.) Specifically, we generate samples of the model defined on a $12 \times 12$ torus over a grid of $(s, J^{-1})$ values and plot the corresponding ferromagnetic order parameter (Fig. 1c), defined as

$$m = \frac{1}{N} \left| \sum_{i=1}^{N} \sigma_i \right|, \tag{4}$$

and the antiferromagnetic order parameter or staggered magnetization (Fig. 1d), defined as

$$m_{\text{AFM}} = \frac{1}{N} \left| \sum_{i=1}^{N} (-1)^{x_i + y_i} \sigma_i \right|. \tag{5}$$

Here, $N$ denotes the number of spins in the system (144 in this case), and $(x_i, y_i)$ denotes the location of the $i^{th}$ spin. Although thermodynamic phase transitions are typically sharp only in the infinite-size limit, finite-size systems exhibit a smooth transition in the order parameter when crossing a critical point. We find that the boundary separating the ferromagnetic phase (order parameter close to 1) from the paramagnetic phase (order parameter close to 0) qualitatively matches the critical line obtained from the exact solution; similarly, good agreement is found for the antiferromagnetic-to-paramagnetic critical line.

Before proceeding, let us highlight a subtle point. We find that a small region corresponding to low temperature (i.e., low $J^{-1}$) and $s > 1$ in Fig. 1d exhibits low staggered magnetization values. This behavior is expected to arise even in classical Monte Carlo simulations due to the phenomenon of entropy-driven ordering[51] that is exhibited by the PUD model. In particular, the order parameter in this regime is expected to saturate only in the infinite size limit, with smaller values observed for finite system sizes, consistent with our QA results. We will explore this exotic behavior in future work[52].

While computing the order parameters for a finite-sized system does provide a qualitative understanding of the location of critical lines, a more robust method involves the computation of Binder cumulants, which we describe below.

## Finite-size scaling and critical exponents

The exact solution indicates that for each value of $s$, the system undergoes a second-order phase transition at a corresponding critical temperature $T_c(s)$. (Exact solution expressions for the critical temperature are provided in the Methods section.) To experimentally identify the critical points as a function of the tuning parameter $s$, we employ two complementary methods: the crossing of Binder cumulants and finite-size scaling (FSS) analysis of susceptibility[53]. We restrict our analysis to the regime $s < 1$ (and thus to the ferromagnetic-paramagnetic transition) and extract critical energy scales (i.e., those corresponding to critical temperatures) specifically at $s = 0, 0.2, 0.4, 0.6$, and $0.8$. (A similar analysis can be implemented for $s > 1$, by replacing the ferromagnetic order parameter $m$ by the antiferromagnetic order parameter $m_{\text{AFM}}$, in the analysis below.) For each value of $s$, to probe different effective temperatures, we consider a uniform grid of 30 $J^{-1}$ values, centered around the anticipated critical value of $J^{-1}$. To minimize boundary effects, we implement sampling only for systems with periodic boundary conditions along both spatial directions. Specifically, we considered toroidal lattices with $N = L \times L$ spins and $L \in \{6, 8, 10, 12\}$, as well as skew-toroidal lattices with $N = (L \times L)/2$ spins and $L \in \{8, 12, 16\}$ (more details are provided in the Methods section). For each system size and each $J^{-1}$ value, we sampled $10,000$ spin configurations and computed the fourth-order Binder

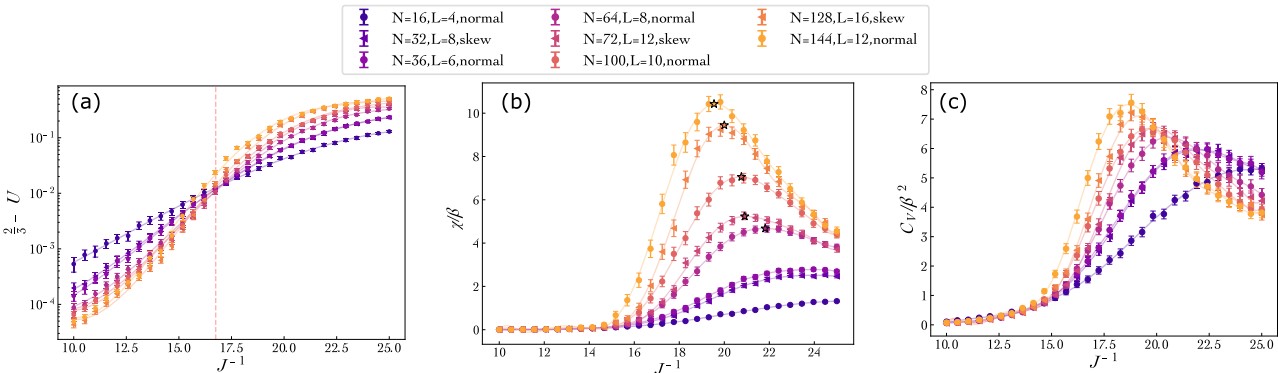

**Fig. 3 | Thermodynamic quantities as a function of inverse energy scale for $s = 0$.** Shown here are **a** the fourth-order Binder cumulants (subtracted from 2/3), **b** magnetization susceptibility times temperature, $\chi/\beta$ and (c) the heat capacity times temperature squared $C_V/\beta^2$ for various system sizes as a function of the inverse energy scale $J^{-1}$. The star symbols in (**b**) denote the peak values of a polynomial fit for each of the different system sizes. The error bars denote one standard deviation values obtained from a bootstrap analysis.

cumulant[53]

$$U = 1 - \frac{\langle m^4 \rangle}{3\langle m^2 \rangle^2}, \qquad (6)$$

where $m$ denotes the magnetization per spin of a configuration. While implementing Monte Carlo simulations, the critical temperature can be determined by identifying the intersection of Binder cumulant curves for various system sizes[26,53]. Analogously, for each value of $s$ considered in our study, we observe a clear crossing of the Binder cumulant curves, enabling us to identify the energy scale $J_c$ corresponding to the critical temperature $T_c$. For example, Fig. 3a shows the Binder cumulant curves for $s = 0$. A slight scatter in intersection points is observed, consistent with finite-size deviations expected to arise from the limited system sizes imposed by hardware constraints. (While the Binder cumulant curves should intersect at a single point in the thermodynamic limit, finite-size corrections can cause noticeable deviations[26,53,54]. We obtain a similar scatter in intersection points using MCMC on systems of similar sizes; see Supplementary Material).

For each value of $s$, we compare the inverse critical energy scale $J_c^{-1}(s)$ extracted from the crossing of Binder cumulant curves in the quantum annealing data, with the critical temperature $T_c(s)$ obtained from the exact solution. Consistent with our initial hypothesis, Eq. (3), the two quantities exhibit a linear relationship (see inset of Fig. 2), implying self-consistency. Using the corresponding linear fit and the analytical expression for the critical line $s(T)$ (see Supplementary Material), we obtain a prediction for the critical inverse energy scale $J_c^{-1}(s)$ as a function of $s$ for $s \in [0, 1]$. When this curve is overlaid on the average magnetization plot for the $12 \times 12$ system size, it delineates the boundary between the ferromagnetic and paramagnetic phases (Fig. 2).

The critical energy scale can also be identified using an FSS analysis of the magnetization susceptibility (per spin) $\chi$ and the heat capacity (per spin) $C_V$ as a function of the system size and energy scale. To that end, we extract $\chi/\beta$ and $C_V/\beta^2$ from the quantum annealing samples, using

$$\frac{\chi}{\beta} = N(\langle m^2 \rangle - \langle m \rangle^2), \qquad (7a)$$

$$\text{and } \frac{C_V}{\beta^2} = \frac{(\langle E^2 \rangle - \langle E \rangle^2)}{N}, \qquad (7b)$$

where $m$ and $E$ denote the magnetization per spin and energy [with respect to Eq. (1a), without any energy scaling] of spin

configurations. We note that since $\beta$ is unknown, $\chi$ and $C_V$ cannot directly be extracted by multiplying $\frac{\chi}{\beta}$ and $\frac{C_V}{\beta^2}$ respectively by appropriate powers of $\beta$. (While $\chi$ was heuristically obtained in Ref. 9 using a time-varying ramp for the longitudinal field, our approach is based on Boltzmann sampling. Hence, using (7a) is more appropriate.) We plot both these quantities as a function of $J^{-1}$ for various system sizes and for various $s$. The obtained trends (see Fig. 3b, c) are consistent with those usually observed in Monte Carlo simulations and finite-size scaling theory[55,56]: As we increase the system size, the heights of the peaks increase, and their corresponding temperatures move towards lower temperatures (which correspond to lower values of $J^{-1}$ in our case).

Next, we implement an FSS analysis of the susceptibility as a function of $J^{-1}$ and system size. We recall that close to a continuous phase transition, the correlation length $\xi$ and the magnetization susceptibility $\chi$ diverge as the reduced temperature $t = (T - T_c)/T_c$ approaches 0. The critical exponents $\nu$ and $\gamma$ are defined through the relationships

$$\xi \propto t^{-\gamma} \text{ and } \chi \propto t^{-\nu}. \qquad (8)$$

At $s = 0$, the exact values of the exponents are known to be $\gamma = 1.75$ and $\nu = 1$. Using the Metropolis Markov-chain Monte Carlo (MCMC) method, we find that the exponents do not change as a function of $s$ for $s \in [0, 1)$ (see Methods section for more details about the MCMC simulations).

We now discuss the procedure for extracting the exponents and critical inverse energy scale using quantum annealing. To that end, we first recall that the values $\gamma(s)$, $\nu(s)$, and $T_c(s)$ can be obtained from the locations and heights of peaks of the $\chi(L, T, s)$ curves as a function of $T$ for various $L$, with $\chi$ extracted from any sampling method (such as QA or MCMC). For notational convenience, we will drop the dependence on $s$. Let $\chi_L$ and $T_L$ denote the height and temperature corresponding to the peak of the curve $\chi(L, T)$ as a function of $T$. The finite-size scaling ansatz implies that[26,27]

$$T_L = T_c(1 + aL^{-1/\nu}), \qquad (9a)$$

$$\text{and } \log \chi_L = b + \gamma/\nu \log L. \qquad (9b)$$

Here, $a$ and $b$ are some constants that are not of interest. While Eqs. 9 can be used to extract $\nu$, $\gamma$ and $T_C$ when using MCMC, it is not directly suitable for our quantum annealing-based sampling method, since we do not have direct access to $T$, or to $\chi$. Since $T \propto J^{-1}$, we expect that the

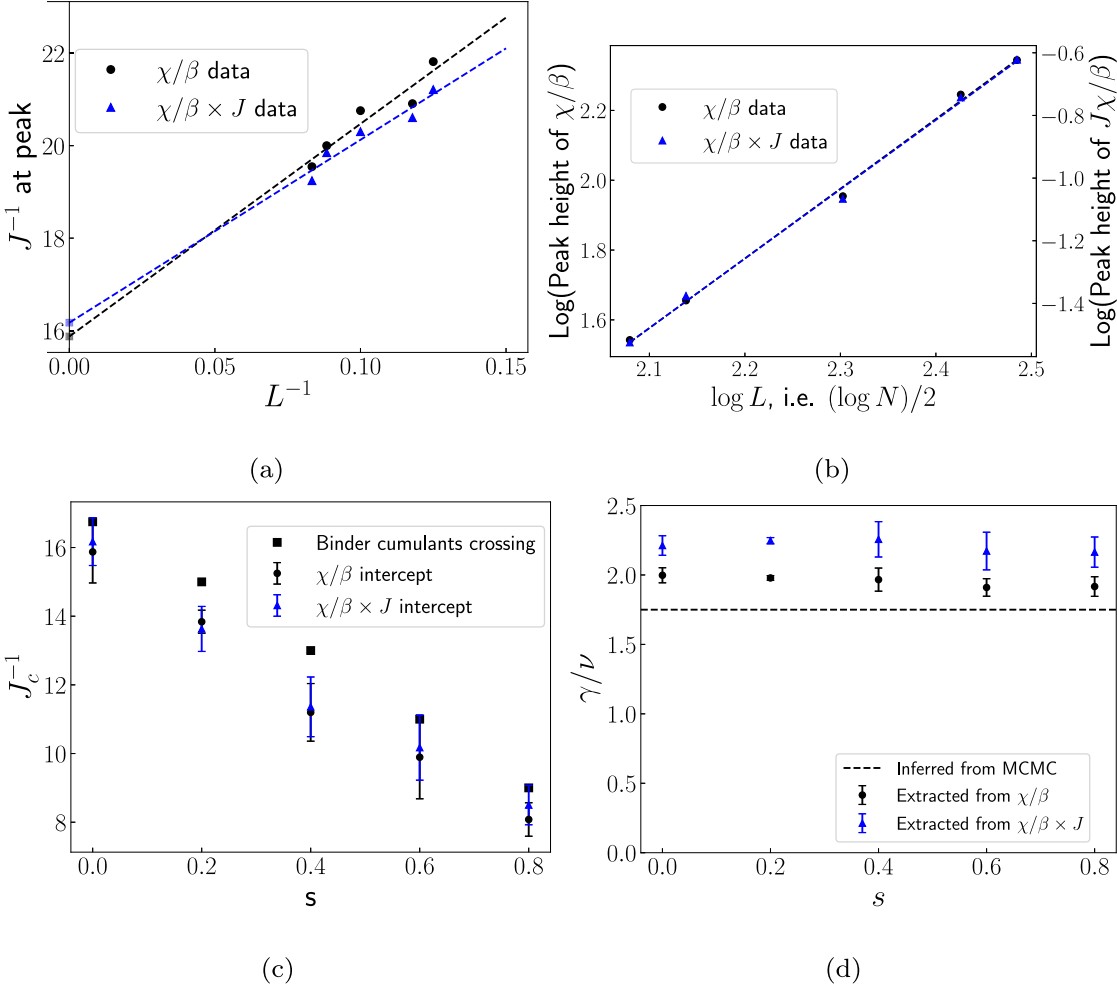

(a)                                                              (b)

(c)                                                              (d)

**Fig. 4 | Finite-size scaling (FSS) analysis on quantum annealer data.** For obtaining the critical inverse energy scale $J_c^{-1}$ and the ratio of exponents $\gamma/\nu$. Plots **a** and **b** correspond to $s = 0$. **a** The locations of peaks of $\chi/\beta$ and $\chi/\beta \times J$ as a function of inverse system size $L^{-1}$. The intercept corresponding to the best linear fit yields $J_c^{-1}$. **b)** The slope of the best linear fit of the logarithm of the peaks of $\chi/\beta$ (and of $\chi/\beta \times J$) vs the logarithm of system size $L$ yields $\gamma/\nu$. **c** The critical inverse energy scale and **d** the extracted values of $\gamma/\nu$ as a function of the interpolation parameter $s$. The 'correct value' (inferred from MCMC), $\gamma/\nu = 1.75$ is plotted as a dashed horizontal line. The error bars denote one standard deviation values obtained from a bootstrap analysis.

susceptibility $\chi$ will be proportional to the product $\frac{\chi}{\beta} \times J$. Hence, we plot $\frac{\chi}{\beta} \times J$ as a function of $J^{-1}$ for various system sizes, to mimic the dependence of $\chi$ as a function of temperature. Let us denote the peak heights and locations by $\chi_L$ and $J_L^{-1}$ respectively. Then, from Eqs. (3) and (9a), it follows that

$$J_L^{-1} = J_c^{-1}(1 + x_0 L^{-1/\nu}). \tag{10}$$

A curve fitting procedure is typically used to simultaneously extract $\nu$ and the critical temperature (instead, $J_c^{-1}$ in our case). While the curve fit procedure yielded $\nu$ between 1.25 to 1.5 for the five $s$ values, the extracted $J_c^{-1}$ values were found to be significantly different (up to a factor of two larger) from those expected from Fig. 1c, suggesting that these results are unreliable. Instead, we assume $\nu = 1$ for all $s$ (validated by MCMC; see Methods section for more details), and plot $J_L^{-1}$ as a function of $L^{-1}$. From Eq. (10) (with $\nu = 1$), the intercept then corresponds to $J_c^{-1}$. The linear fit for $s = 0$ is shown in Fig. 4a. The vertical axis intercept yields our estimate for $J_c^{-1}$ at $s = 0$. While these values differ from those obtained using crossings of Binder cumulants, both data sets follow similar trends as a function of $s$. (Finite-size effects account for the small mismatch between the two $J_c^{-1}$ estimates.)

Turning to Eq. (9b), we note that the slope in the linear relationship between $\log \chi_L$ and $\log L$ corresponds to $\gamma/\nu$. The extracted

values of $\gamma/\beta$ and their associated errors (equal to the standard deviation of the slope in the linear fits) are plotted as a function of $s$ in Fig. 4d. The values are consistently seen to lie around 2.2, which is off by approximately 25% from the correct value of 1.75 (inferred from MCMC). This discrepancy may stem from deviations from the hypothesized relation, Eq. (3), which was applied at two stages in the analysis, in addition to the relatively small system sizes used in our analysis. Understanding the relative contributions of these factors would require implementing alternative temperature estimation techniques, and access to a larger range of system sizes.

To avoid these issues, we propose and implement an alternate procedure, based on directly using the peak locations and heights of $\chi/\beta$ (without multiplying by $J$). This avoids the second potential source of error, i.e., the application of the temperature-inverse energy scale relationship. Specifically, we propose that the values $\chi_L$, $T_L$, and hence $J_L^{-1}$ in Eqs. (9) and Eq. (10) may be replaced by the peaks of $\chi/\beta$ instead, to obtain $\gamma/\nu$ and critical values of inverse energy scales. We base this on the difference between the peak locations (peak heights) of the two quantities being infinitesimally small (related by a constant) in the limit of infinite system sizes. (See Supplementary Information for an argument.) While the system sizes in our analysis are fairly small, we find the extracted values (obtained from this alternate procedure) of $\gamma/\nu$ to be

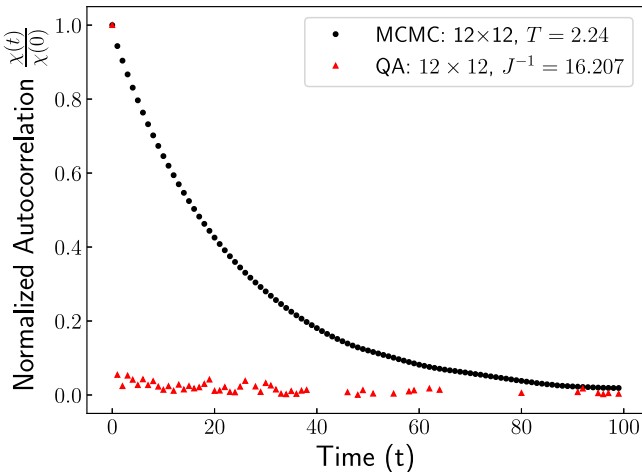

**Fig. 5 | Absence of critical slowing down using quantum annealing.** Normalized autocorrelation function $\chi(t)/\chi(0)$ as a function of time $t$ for Markov-chain Monte Carlo (MCMC) simulations and for quantum annealing samples. Plotted data corresponds to $s = 0.0$. The temperature and inverse energy scale were chosen to be close to their respective critical values.

around 2 for various values of $s$, thus being closer to the correct answer of 1.75 (see Fig. 4d).

In contrast to the order parameters, Eqs. (4) and (5) discussed above, the susceptibility, heat capacity, and Binder cumulants require the computation of up to fourth-order moments of magnetization and energy, and are thus sensitive to experimental errors. We find that a calibration-refinement technique is crucial to obtain reliable results[57].

Without calibration refinement, the device noise in the relevant parameter regimes manifests as disorder in both the longitudinal field and coupling strengths, with each anneal effectively sampling from a different disorder realization[40]. The presence of such disorder can alter the critical exponents relative to those of the clean (i.e., disorder-free) model, particularly when the heat capacity exponent $\alpha$ for the clean system is greater than zero[58]. In the 2D Ising model (and the PUD model we consider), with $\alpha = 0$, weak uncorrelated disorder is believed not to change the critical exponents, though it may introduce a logarithmic correction to scaling.

In the Supplementary Material, we provide a detailed description of the calibration refinement procedure we use as well as a before/after comparison for various physical quantities.

## QA circumvents critical slowing down

Our work addresses a persistent challenge in computational statistical mechanics. One of the key challenges faced by many Markov-Chain Monte Carlo (MCMC) methods, such as the single-spin flip algorithm, is the phenomenon of critical slowing down near continuous phase transitions[27,59]. In this regime, the system's convergence to equilibrium becomes significantly slower, and many updates are required to generate statistically independent samples. This issue is particularly severe for certain frustrated systems such as spin-glasses or spin ices. While model-specific Monte Carlo algorithms incorporating cooperative updates can partially alleviate the problem[26], no universal strategy has been found to eliminate it across arbitrary models.

Our QA-based approach naturally evades this issue. Each sample in QA is generated by initializing the qubits in the all-up state with respect to the Pauli $X$ basis, evolving them with a time-evolving Hamiltonian, and finally measuring in the Pauli $Z$ basis. Since the initialization operation is identical for each sample (and independent of the measurement output of any prior sample), we expect our samples to be statistically independent by design. Any residual

correlations, if present, would primarily arise from device-level non-idealities rather than the sampling procedure itself.

While critical slowing becomes increasingly significant for large system sizes, our results show that quantum annealing largely circumvents this issue. To directly compare slowing down for our single-spin-flip Metropolis implementation and our quantum annealing approach, we choose the largest system size we used for QA, which is $N = 12 \times 12$ spins. For a fair comparison, we implement MCMC simulations for the same system size. To measure the degree of correlations between samples that are $t$ steps apart, we compute the autocorrelation function $\chi(t)$ using[27]

$$
\chi(t) = \frac{1}{t_{\max} - t} \sum_{t'=0}^{t_{\max}-t} m(t')m(t'+t)
$$
$$
- \frac{1}{t_{\max} - t} \sum_{t'=0}^{t_{\max}-t} m(t') \times \frac{1}{t_{\max} - t} \sum_{t'=0}^{t_{\max}-t} m(t'+t).
$$
(11)

where $t_{\max}$ is the total number of samples or steps generated by the algorithm. (For the quantum annealing simulations, we generated $t_{\max} = 10,000$ samples.) For the MCMC simulations, following standard convention, one unit of time is taken to correspond to one MCMC update per spin in the system.

In Fig. 5, we plot the normalized autocorrelation function for QA, along with MCMC simulations for $s = 0$ close to the critical point. The MCMC data exhibits the expected exponential decay in autocorrelation, indicative of correlations persisting over many time steps, especially close to criticality. In contrast, the QA result shows a small, approximately constant autocorrelation value, suggesting little to no temporal correlation between samples. In typical MCMC simulations, the normalized autocorrelation function decays exponentially with a characteristic timescale $T_A$, which diverges with increasing system size near the critical point[27]. The absence of such behavior in the QA results-particularly the lack of any discernible correlation timescale even near the critical point-supports the conclusion that QA effectively bypasses critical slowing down.

To further investigate correlations near the phase transitions, we examine the average value of the normalized autocorrelation function and again observe no significant increase in its magnitude close to the transition points. (See Supplementary Material for more details).

We note that the critical slowing down discussed here pertains specifically to the generation of classical samples, whether via classical MCMC or using QA, used to study a classical model (such as the PUD model). In the context of QA, a distinct form of slowing down also arises: the time-dependent Hamiltonian of the QA device typically traverses a quantum phase transition, near which the quantum dynamics of the qubits can slow down or freeze[59]. Related phenomena such as the Kibble-Zurek scaling of defects have been probed in quantum annealers[13,15,17]. However, our focus here is not on this quantum slowing down, but rather on the behavior of QA viewed as a sampling algorithm for classical statistical physics models.

## Discussion

We have shown that quantum annealing can quantitatively capture finite-temperature critical phenomena in classical systems with tunable frustration. We demonstrated that quantum annealers can be used to identify critical points and extract universal exponents-key signatures of criticality that are often difficult to access using classical Monte Carlo methods.

This work establishes a new methodological benchmark: by applying finite-size scaling and Binder cumulant analysis directly on hardware, we confirm that annealer output closely approximates thermal distributions, yielding consistent critical behavior and scaling collapse.

The model's tunable frustration provides a stringent test, and the ability to reproduce universality class behavior in this setting addresses long-standing doubts about QA's suitability as a thermal sampler, especially in frustrated settings. Crucially, we observe no signs of critical slowing down, suggesting a potential computational advantage over classical approaches near second-order transitions.

Generating configurations from the thermal distribution of an arbitrary Ising model is a challenging problem. Even for structured Ising models that represent real materials or idealized theoretical systems, such sampling becomes particularly difficult near critical points. Beyond critical phenomena, obtaining high-quality samples can also be challenging in other regimes, such as at low temperatures or in systems with large degeneracies, including spin glasses. The lack of critical slowing down and partial success in extracting critical exponents in a model with geometric frustration thus positions quantum annealers as emerging tools for sophisticated studies of the thermodynamics of complex systems.

Nonetheless, we find that careful calibration refinement, while potentially a computational bottleneck, is crucial for obtaining reliable results. In addition, accurate estimation of critical exponents requires studying large system sizes, which can be limited by the number of qubits and their connectivity in the QPU. We anticipate that continued improvements in quantum hardware, such as noise reduction and an increase in the qubit number and connectivity, will make quantum sampling methods a practical alternative for studying equilibrium criticality and other thermodynamic properties in models that are classically costly or inaccessible. We expect our conclusions to extend beyond the PUD model. Investigating the performance of quantum annealers for studying other types of phase transitions, such as discontinuous ones or Berezinskii-Kosterlitz-Thouless phase transitions, would be an interesting area for future research.

Following the release of the first version of our preprint, another study appeared showcasing precise extraction of critical exponents for the 2D Ising model[60].

## Methods
### Quantum annealing protocol
For all our quantum annealing (QA) experiments reported in the main text, we used D-Wave's `Advantage2_prototype2.6` device, accessed via the Leap cloud-computing service provided by D-Wave Quantum Inc.[61]. We used an annealing time of $T_A = 100\mu s$ for each experiment. To compute the ferromagnetic and antiferromagnetic order parameters reported in Fig. 1, we obtained 10, 000 samples, with a programmed time gap known as "readout thermalization", of 10, 000$\mu s$ to allow the hardware to cool down after each sampling experiment and to improve result quality. We used 1000 bootstrap resamples with 1000 samples each in order to compute the mean and standard deviation of our estimates of the two order parameters, and found the errors to be negligible for all considered values of $s$ and $J^{-1}$.

We extracted the inverse critical energy scales $J_c^{-1}$ and the ratio $\gamma/\nu$ for a grid of interpolation parameter values $s \in \{0, 0.2, 0.4, 0.6, 0.8\}$. For each $s$ value, $J_c^{-1}$ was obtained by identifying the intersection of Binder cumulants as well as by using finite-size scaling (FSS). For each $s$, we implemented sampling experiments for many system sizes and for a range of energy scale values. We implement these experiments for two types of systems: (i) normal toroidal systems with a number of spins $N = L \times L$, and (ii) skew toroidal systems with $N = (L \times L)/2$, with $L$ being an integer. Further information regarding the embeddings and spin-reversal/gauge transformations implemented for the experiments is provided in the Supplementary Material.

### Calibration refinement
To reduce the effects of non-idealities in quantum annealing, such as crosstalk, device variation, and environmental noise, we implement a calibration refinement protocol, which is also known as "shimming"[57].

This procedure is often crucial to obtain reliable results, and has been used in many previous works that have leveraged quantum annealing to study condensed matter physics models[10,13,18]. We implemented shimming to extract the crossings of the Binder cumulant and to extract the value of $\gamma/\nu$.

We implemented calibration refinement of the couplers and flux bias offsets (FBOs) with an adaptive step size. In addition, we implemented a smoothing procedure for FBOs and coupler shims across various energy scales. Further details are provided in the Supplementary material.

### Monte Carlo Simulations
To obtain the ground truth for the critical exponents of the PUD model, we implemented single spin flip Markov Chain Monte Carlo (MCMC) algorithms for a range of system sizes at $s = 0, 0.2, 0.4, 0.6$ and 0.8. Details regarding the parameter values used and the data collapse method used for critical exponent extraction are provided in the supplementary material.

## Data availability
The data generated in this study have been deposited at https://github.com/PratikSathe/PUD_classical_criticality.

## Code availability
The calibration refinement code used in this work is a modified version of the code accompanying Ref. 60. The modifications and parameters are described in the supplementary material.

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

## Acknowledgements

We thank Cristian Batista for helpful discussions. The authors acknowledge the support of NNSA for the U.S. DOE at LANL under Contract No. DE-AC52-06NA25396, and Laboratory Directed Research and Development (LDRD) for support through 20240032DR. Research presented in this article was also supported by the National Security Education Center (NSEC) Information Science and Technology Institute (ISTI) using the Laboratory Directed Research and Development program of Los Alamos National Laboratory project number 20240479CR-IST. PS also acknowledges the support via the ISTI Fellowship. Assigned: Los Alamos Unclassified Report LA-UR-25-23528. LANL is managed by Triad National Security, LLC, for the National Nuclear Security Administration of the U.S. DOE under contract 89233218CNA000001. LA-UR-25-23528. We would also like to thank the New Mexico Consortium, under subcontract

C2778, the Quantum Cloud Access Project (QCAP), for providing quantum computing resources and technical collaboration.

## Author contributions

F.C. and P.S. conceived the initial experiment, and P.S. subsequently expanded the scope of the project. P.S., A.K. and S.M. performed the quantum annealer experiments. P.S. implemented the supporting classical Monte Carlo simulations. P.S. and F.C. wrote the initial manuscript. All authors discussed the results and contributed to the final version of the manuscript. C.N., C.C. and F.C. provided supervision.

## Competing interests

PS is an employee of D-Wave Quantum Inc., and ADK is an employee and stockholder; both authors declare competing interests on that basis. The remaining authors declare no competing interests.
