## [Transparent Peer Review file · Nature Communications]

Classical Criticality via Quantum Annealing

Corresponding Author: Dr Pratik Sathe

Version 0:

Reviewer comments:

Reviewer #1

(Remarks to the Author)

This manuscript provides an excellent demonstration of applying quantum-computing techniques and resources to the study of physical systems. The authors formalize and demonstrate how quantum annealing techniques can be utilized to systematically study finite-temperature statistical physics in geometrically frustrated spin models, a topic with recent empirical evidence but with, until now, weaker theoretical rigor and classical computational evidence. Overall, the manuscript is well written and will be of immediate interest to researchers broadly in the areas of quantum computing, computational and statistical physics, and classical computer science.

With that said, I do have some questions/suggestions that should be addressed within the manuscript to enhance the clarity and impact of the work. I list a few notable questions/comments here, but I provide (attached) an annotated manuscript with specific highlighted areas and comments/questions specific to each.

One first question is about the motivation for selecting the Piled-Up Dominoes (PUD) model. Does the PUD model have real-world applications or is it connected to a relevant class of materials/engineering/optimization problems? I think it is made clear that the PUD model is a useful toy problem, but some discussion should be provided to how it connects to real-world challenges and how representative this problem's behavior is for relevant problems. Addressing this will help emphasize the impact and relevance of the work.

Next, it appears (based on the figure 1d) that a fourth phase is observed in the antiferromagnetic model at $s > 1$ and $1/J < 1$ (the upper left-hand corner). What is the origin of this? This discrepancy between the theoretical and quantum-annealing results are not addressed.

Another point that should be addressed is discussion about uncertainties and noise in the quantum annealing experiments. It is mentioned a few times (see highlighted manuscript) that some quantities either could not be estimated or were not reliable because the data was "too noisy." Why is this the case? Is it limited, e.g., by hardware noise, by finite sample numbers, or by quantum effects? Also, how does this uncertainty propagate to the uncertainty of the estimated critical quantities? Is this specific to the PUD problem?

Finally, a primary conclusion of the work is that quantum annealers could be useful tools for study of thermodynamic physics. Given the broad readership of the journal, I think it beneficial to provide a few more sentences of discussion around research opportunities and references that could guide interested readers in the frontier of this area.

Reviewer #2

(Remarks to the Author)

I have read the manuscript by Sathe et al., entitled "Classical Criticality via Quantum Annealing". They describe a transformative application of quantum annealing (QA) in classical statistical physics. The work is compelling and I recommend it to be published, if the following questions and concerns can be addressed:

Comment 1:

The caption of Figure 1 states that the data "are in agreement with the exact solution." Would it be more appropriate to phrase this as "qualitatively matches" instead? This adjustment would better reflect the subsequent quantitative analyses.

Comment 2:

Figure 2 references the fourth-order Binder cumulant U (Eq. 6) and its crossing points to extract J_c . To improve readability, consider relocating this validation to Section II (after introducing finite-size scaling formally).

Comment 3:

The manuscript notes: 'A slight scatter in intersection points is observed, consistent with finite-size deviations expected to arise from the limited system sizes imposed by hardware constraints.' Can you explain more about this? The trends in Fig. 2 suggest the line with larger system sizes do not necessarily converge toward J_c . Is this interpretation accurate?

Comment 4:

I notice that calibration refinement is essential for reliably measuring the quantities like magnetization susceptibility (Appendix A.3). This process reduce the effects of environmental noise—such as D-Wave's intrinsic ~ 10 mK thermal noise—by adjusting flux-bias offsets (FBOs) and coupler strengths. Could this suppression artificially reduce thermal fluctuations critical to critical phenomena, potentially explaining why the QA-measured critical exponent ratio γ/ν deviates by $\sim 25\%$ from the theoretical value of 1.75 (Fig. 4d)?

Reviewer #3

(Remarks to the Author)

This manuscript presents the first demonstration of directly analyzing finite-temperature critical phenomena in a classical statistical model using quantum annealing (QA) hardware, representing a significant advance in the field. By tuning the Hamiltonian's energy scale to control the effective temperature, the authors apply Binder cumulant analysis and finite-size scaling (FSS) to the Piled-Up Dominoes (PUD) model. This approach successfully reconstructs the phase diagram and estimates critical exponents, while also providing quantitative evidence that QA sampling exhibits little to no critical slowing down, a pronounced issue in classical Monte Carlo methods.

The work holds substantial significance for both statistical physics and quantum computing. While previous studies (e.g., Harris et al., *Science* 361, 162 (2018); King et al., *Nat. Commun.* 12, 1113 (2021)) have explored criticality with QA under specific conditions, this paper addresses a more challenging and tunably frustrated system, achieving hardware-based temperature control and phase diagram reconstruction. This sets a new benchmark for demonstrating the robustness and potential generality of the methodology. Moreover, the findings suggest that QA could circumvent computational bottlenecks encountered in classical methods near criticality, paving the way for broader applications.

The main claims, effective temperature control, phase diagram reconstruction, critical exponent extraction, and avoidance of critical slowing down, are generally supported by the presented data. However, the observed deviation of up to 25% in γ/ν from the theoretical value of 1.75 warrants further investigation to disentangle contributions from the energy scale–temperature approximation and finite-size effects. Demonstrating applicability to other models or types of phase transitions, such as Berezinskii–Kosterlitz–Thouless or first-order transitions, would further strengthen the generality of the conclusions, if the authors can.

There are a few aspects of the data analysis and interpretation that could be improved. Notably, the error sources in critical exponent estimation have not been quantitatively separated, and no numerical before/after comparison of calibration refinement (shimming) is provided. While these issues do not constitute fatal flaws that preclude publication, clarifying them would enhance the paper's credibility.

The methodology is sound, with established statistical physics techniques such as Binder cumulant crossings and FSS applied appropriately. QA-specific procedures, including temperature tuning and embedding strategies, are reasonable and meet reproducibility standards in the field. The experimental conditions, sample sizes, system configurations, energy scale settings, and calibration steps (including fixed gauge transformation) are described in sufficient detail to enable replication. Providing more explicit numerical parameters or code snippets for the shimming procedure in the Supplementary Material would make replication even more straightforward.

Overall, this manuscript exhibits high originality and scholarly value. With the suggested clarifications and minor enhancements, it would be a strong candidate for publication in *Nature Communications*.

Version 1:

Reviewer comments:

Reviewer #2

(Remarks to the Author)

I have read through the revised manuscript and the authors' response to my comments. The authors have adequately addressed my concerns, particularly by including the further analysis on various error sources and the efficacy of the calibration refinement procedure. These additions, including the before/after comparison of key physical quantities, have made the data more reliable and the results more reproducible. The manuscript is significantly improved over the initial version.

I therefore recommend the work for publication.

Reviewer #3

(Remarks to the Author)

The authors have responded sincerely to all of reviewers' comments. I find their replies appropriate and satisfactory, and I have no further concerns.

Response to the Referees of “Classical Criticality via Quantum Annealing”

Pratik Sathe, Andrew D. King, Susan M. Mniszewski,
Carleton Coffrin, Cristiano Nisoli, Francesco Caravelli

Oct 17, 2025

We are immensely grateful to the reviewers for the appreciation of our work, and even more so because of their very useful suggestions.

We have made modifications to the revised manuscript to incorporate the referee suggestions. In particular, we have expanded on the discussion about the sources of errors in our computation of the critical exponents, and have analyzed the impact of calibration refinement.

Below, we show the referee comments in black followed by our responses in blue. Changes made in the manuscript, when quoted here, appear in red font. The reference numbers in the bibliography of this document are different from those in the updated manuscript.

1 Reviewer 1

This manuscript provides an excellent demonstration of applying quantum-computing techniques and resources to the study physical systems. The authors formalize and demonstrate how quantum annealing techniques can be utilized to systematically study finite-temperature statistical physics in geometrically frustrated spin models, a topic with recent empirical evidence but with, until now, weaker theoretical rigor and classical computational evidence. Overall, the manuscript is well written and will be of immediate interest to researchers broadly in the areas of quantum computing, computational and statistical physics, and classical computer science.

We thank the reviewer for the appreciation of our work.

With that said, I do have some questions/suggestions that should be addressed within the manuscript to enhance the clarity and impact of the work. I list a few notable questions/comments here, but I provide (attached) an annotated manuscript with specific highlighted areas and comments/questions specific to each.

1. One first question is about the motivation for selecting the Piled-Up Dominoes (PUD) model. Does the PUD model have real-world applications or is it connected to a relevant class of materials/engineering/optimization problems? I think it is made clear that the PUD model is a useful toy problem, but some discussion should be provided to how it connects to real-world challenges and how representative this problem's behavior is for relevant problems. Addressing this will help emphasize the impact and relevance of the work.

We thank the referee for this comment. Various considerations motivate the choice of the PUD model. As an exactly solved model, it is well suited for benchmarking; as a simple Ising model on a square grid, it directly and easily embeds onto current QPUs. Importantly, its tunable degree of frustration allows for three distinct thermodynamic phases. While the $s = 0$ limit corresponds to the 2D Ising model, which is probably the first exactly solved model in statistical physics, the $s = 1$ limit shows the complexity of frustrated constrained disorder (Villain, 1977), of a topological nature (it can be mapped to a dimer model). Furthermore, it inspired Villain et al.'s early studies of the the order-by-disorder phenomenon (Villain et al., 1980).

We have revised the manuscript to clarify these motivations and to include this historical context alongside the benchmarking rationale, by adding the following.

In addition to its rich phenomenology, this model played a key role in the theoretical development of the order-by-disorder phenomenon by inspiring Villain et al.'s domino model [1]. Moreover, its Ising form, regular lattice structure, and exact analytical solution make it an ideal testbed for benchmarking quantum annealers. Geometrically frustrated systems are generally more challenging to simulate numerically [2], and the tunable degree of frustration in the PUD model controls this sampling difficulty. The competing and mutually-incompatible interactions that define geometric frustration can give rise to exotic phases such as spin ice and spin liquid phases [3]. The PUD model therefore serves as both a meaningful and demanding benchmark for assessing the effectiveness of our approach.

2. Next, it appears (based on the figure 1d) that a fourth phase is observed in the antiferromagnetic model at $s > 1$ and $1/J < \sim 1$ (the upper left-hand corner). What is the origin of this? This discrepancy between the theoretical and quantum-annealing results are not addressed.

We congratulate the referee for noticing a tiny detail that hides very important physics. That little sliver at small T and $s > 1$ is in fact not a third phase, but it comes from an exotic physical phenomenon called “entropy driven ordering”—which is distinct from “order by disorder”. In this case, the system orders, paradoxically, to increase the entropy of the ground state.

To explain: The $s > 1$ region at very low temperature is still in the anti-ferromagnetic phase (therefore it is not a fourth phase). But contrary to standard energy-driven ordering, in entropy-driven ordering the order parameter approaches maximal value (here 0.25) as the system approaches the thermodynamic limit (in regular ordering, instead, a finite size system orders more easily). This is because the order emerges to maximize entropy. There is a competition between a bulk entropy gain, that comes from ordering the ferromagnetic columns and is extensive, and a sub-extensive entropy loss from column ordering: only in the thermodynamic limit one fully wins over the other. Something similar has been observed and reported experimentally in artificial nanomagnet systems by one of our coauthors (see [4] for a discussion). What we observe is therefore expected, and consistent with this phenomenon. Had we used a very large system, that sliver would not have been noticeable. Of course that would have run contrary to our use of finite size scaling.

While this issue does not align to the aims of this manuscript, it is most interesting and will be addressed

in a future work. Nonetheless, we also agree that a brief discussion is warranted here. We added:

Before proceeding, let us highlight a subtle point. We find that a small region corresponding to low temperature (i.e., low J^{-1}) and $s > 1$ in Fig. 1d exhibits low staggered magnetization values. This behavior is expected to arise even in classical Monte Carlo simulations due to the phenomenon of entropy-driven ordering [4] that is exhibited by the PUD model. In particular, the order parameter in this regime is expected to saturate only in the infinite size limit, with smaller values observed for finite system sizes, consistent with our QA results. We will explore this exotic behavior in future work [5].

3. Another point that should be addressed is discussion about uncertainties and noise in the quantum annealing experiments. It is mentioned a few times (see highlighted manuscript) that some quantities either could not be estimated or were not reliable because the data was “too noisy.” Why is this the case? Is it limited, e.g., by hardware noise, by finite sample numbers, or by quantum effects? Also, how does this uncertainty propagate to the uncertainty of the estimated critical quantities? Is this specific to the PUD problem?

We agree with the referee’s comment and have now included further analysis on the various sources of error and the efficacy of the calibration refinement procedure we used to reduce the noise.

(1) In the Methods section, we have now included a before-after comparison of the various physical quantities and statistics (Binder cumulant, χ , C_V) with respect to the calibration refinement procedure. (See Fig. 7 in the updated manuscript).

(2) To confirm that the calibration refinement procedure is indeed improving the solution quality, we show that the magnetization and frustration probability values have lower variance (as desired), after the calibration refinement technique is implemented. (See Fig. 8 in the updated manuscript.)

(3) We confirm that increasing the number of samples is unlikely to change the extracted values of the Binder cumulant, heat capacity and magnetization susceptibility. (See Fig 9 in the updated manuscript.) The same conclusions were found for (staggered) magnetization as well.

While we were unable to identify the precise reason for some of the discrepancies (such as errors in the extracted exponent values), our analysis thus indicates the results should improve with implementations on larger system sizes and improved calibration refinement. We expect that similar issues (and our other conclusions more generally) will apply to other models beyond the PUD model, which we now also state in the Discussions section of the revised manuscript.

4. Finally, a primary conclusion of the work is that quantum annealers could be useful tools for study of thermodynamic physics. Given the broad readership of the journal, I think it beneficial to provide a few more sentences of discussion around research opportunities and references that could guide interested readers in the frontier of this area.

We thank the referee for this suggestion. We have now elaborated on this point in the last two paragraphs in the Discussion section:

“Generating configurations from the thermal distribution of an arbitrary Ising model is a challenging problem. Even for structured Ising models that represent real materials or idealized theoretical systems, such sampling becomes particularly difficult near critical points. Beyond critical phenomena, obtaining high-quality samples can also be challenging in other regimes, such as at low temperatures or in systems with large degeneracies, including spin glasses. The lack of critical slowing down and partial success in extracting critical exponents in a model with geometric frustration thus position quantum annealers as emerging tools for sophisticated studies of the thermodynamics of complex systems. ”

“...Investigating the performance of quantum annealers for studying other types of phase transitions, such as discontinuous ones or Berezinskii–Kosterlitz–Thouless phase transitions would be an interesting area for future research.”

Questions and comments from 1_reviewer_attachment_2_1755210390_convrt.pdf from the decision email.

- FIG. 1d: Why is there an area of low staggered magnetization near 0 on the x-axis? This suggests that the hardware hosts a different phase in that limit? This is not shown to exist on the theory subfigure (b). Our response to a previous comment [“Next it appears (based on figure 1d)...”] has addressed this comment. Furthermore, we have added a discussion (mentioned above) that clarifies this point in the updated manuscript.

- Page 2, col 1: “The PUD model thus provides a meaningful and challenging benchmark for assessing the effectiveness of our approach.” Can this be elaborated on? What real-world applications does the PUD model compare to? Has the computational task of determining thermodynamic properties been assessed? A lot of connections with spin systems and materials have been made but they don’t seem to connect to PUD at this point.

We thank the referee for raising this point. The PUD model can only describe a real material in the same sense as 2D Ising model can. As a model, however, it has a lot of properties that make it a good candidate for benchmarking the performance of a quantum annealing device, which we elaborated on in our response to a previous comment by the referee (“One question is about the motivation for selecting the Piled-Up Dominos model...”). While the difficulties in numerically studying phase transitions in general are well documented in the literature, a study specifically focusing on the computational difficulties for the PUD model is lacking in the literature, to the best of our knowledge. Our response to the previous comment also addresses most of the other points raised in this comment.

- Page 2, col 2: “The model therefore has a tunable frustration.” What is meant by “solved exactly” in this context? Can all the thermodynamic/critical parameters be found analytically? Or just the order parameter?

We thank the referee for raising this point. Following standard terminology in statistical physics, by exact solution, we mean an expression for partition function at finite size or in the thermodynamic limit. Here, the exact solution without an external longitudinal field is known. Expressions for the boundaries separating the three phases have been obtained from the partition function expression. We now state this explicitly in the updated manuscript:

Specifically, expressions for the partition function with zero longitudinal field, and the boundaries separating the three phases have been analytically derived. (The critical exponents have been analytically computed only at $s = 0$.)

- Page 2, col 2: “To control the sampling temperature, we consider a simple model of D-Wave’s QA devices in which the user inputs a Hamiltonian H_{input} , and the device samples from the corresponding Boltzmann distribution at a fixed, but unknown, device-dependent temperature T_{sampler} .” Is this dependent on annealing schedule and annealing parameters?

The sampling temperature typically does depend on the anneal schedule and annealing parameters. We have now added the following clarification, as a footnote [53]:

[53] Since T_{sampler} typically depends on the anneal time and schedule, we fix these to be 100 μs and standard forward anneals throughout our study.

- FIG. 2 caption: What does the color of each curve represent for the Binder cumulants?

The different colors (for the curves) denote different system sizes. We have state this explicitly in the figure caption to avoid confusion with the colorbar values.

- Page 3, col 2: “Using D-Wave’s Advantage2_prototype2.6 quantum annealer, we successfully observe all three phases.” Do you observe a fourth phase near low inverse temperature on the antiferromagnetic model? (See Fig 1d)

We believe that our response to a previous comment (“Why is there an area of low staggered magnetization near 0 on the x-axis...”) addresses this point as well. In summary, a fourth phase is absent, and is instead a consequence of entropy-driven ordering. The new text in the second to the last paragraph just before the beginning of Section II in the revised manuscript clarifies this point.

- Page 5, col 2: “As we increase the system size, the heights of the peaks increase, and their corresponding temperatures move towards lower temperatures (which correspond to lower values of J^{-1} in our case).” Is there a reference for the “usually observed” results in classical MCMC simulations? Can you show the classical MCMC results side by side and see the same finite-size behavior?

We appreciate the referee’s suggestion. We now cite two standard references, namely, the chapter by M. N. Barber, which appears in volume 8 of the collection “Phase transitions and critical phenomena” edited by C. Domb and J. L. Lebowitz, and the book by Cardy, to direct the interested reader to the relevant details. Since our observations follow the established trends of increase in the peak height and decrease in the distance of the temperature at the peak to the critical temperature, we believe that an explicit reproduction of classical MCMC data would not provide additional insight beyond confirming these already well-known results.

- Page 6, col 1: ... our quantum-annealing data was too noisy to accurately extract ν . How could this be fixed? More data? Where is the noise floor?

We thank the referee for raising this important point. Accurate extraction of critical exponents generally requires getting samples at large system sizes, that are much larger than the largest system size (12×12) we were able to simulate using QA. This is a particularly pronounced issue for the exponent ν , which relates to the correlation length changes at the critical point. We believe that the restriction to finite system sizes in our study underlies our inability to accurately extract ν . Simulations of larger system sizes should enable us to extract ν . We expect to be able to do this with access to QPUs with more qubits and with greater connectivity. We have now added a short clarification in the revised manuscript and made the description more accurate:

A curve fitting procedure is typically used to simultaneously extract ν and the critical temperature (instead, J_c^{-1} in our case). While the curve fit procedure yielded ν between 1.25 to 1.5 for the five s values, the extracted J_c^{-1} values were found to be significantly different (up to a factor of two larger) from those expected from Fig. 1c, suggesting that these results are unreliable. Instead, we assume $\nu = 1$ for all s (validated by MCMC; see Methods section for more details), and plot J_L^{-1} as a function of L^{-1} . From Eq. (10) (with $\nu = 1$), the intercept then corresponds to J_c^{-1} . The linear fit for $s = 0$ is shown in Fig. 3a. The vertical axis intercept yields our estimate for J_c^{-1} at $s = 0$. While these values differ from those obtained using crossings of Binder cumulants, both data sets follow similar trends as a function of s .

- Page 6, col 2: “While these values differ from those obtained using crossings of Binder cumulants, both data sets follow similar trends as a function of s .” Is this to be expected? Should Binder cumulants provide a different result? If so, how significant should the difference be and which is more accurate?

As hinted by the referee, one expects that the values of the critical temperature extracted from the two methods should coincide. The discrepancy between the two in our case arises due to the relatively small system sizes we study. We have now added the following clarification in the text:

(Finite-size effects account for the small mismatch between the two J_c^{-1} estimates.)

- Page 6, col 2: “... fourth-order moments of magnetization and energy, and are thus sensitive to experimental errors.” It may be useful to see how your quantitative estimates of the critical parameters change as a function of sample number.

We agree with the referee that this analysis can help in discerning whether increasing the number of samples could improve the accuracy of our estimates. In the Methods section, we have now included an analysis of the effects of number of samples on the estimates of the Binder cumulant, χ/β and C_V/β^2 . In particular, Fig. 9 in the revised manuscript shows these trends for the 12×12 toroidal system. We conclude that the main contributor to the sensitivity of these quantities are experimental inaccuracies, as opposed to insufficient samples or measurements.

- Page 7, col 2: “More broadly, our results position quantum annealers as emerging tools for sophisticated studies of the thermodynamics of complex systems.” I think there should be discussion about the pitfalls of this technique relative to classical methods. For example, if highly accurate estimates are required, how many hardware measurements are required? If you propose QA as a new tool it would be useful to at least briefly discuss where it might be most useful in modern/future workflows.

We appreciate the referee’s comment. While our outlook for QA’s potential is optimistic, certain issues, such as the reliance on calibration refinement and access to small system sizes pose challenges for accurate critical exponent extraction. In the last paragraph of the Discussion section, we have now included a discussion of these challenges:

Nonetheless, we find that careful calibration refinement, while potentially a computational bottleneck, is crucial for obtaining reliable results. In addition, accurate estimation of critical exponents requires studying large system sizes, which can be limited by number of qubits and their connectivity in the QPU. We anticipate that continued improvements in quantum hardware, such as noise reduction and increase in the qubit number and connectivity, will make quantum sampling methods a practical alternative for studying equilibrium criticality and other thermodynamic properties in models that are classically costly or inaccessible.

We have also added a short discussion on natural and interesting future work :

Investigating the performance of quantum annealers for studying other types of phase transitions, such as discontinuous ones or Berezinskii–Kosterlitz–Thouless phase transitions would be an interesting area for future research.

This concludes our response to the referee’s comments. We would like to thank the referee for their time and thoughtful inputs which have helped improve the clarity and the content of our manuscript.

2 Reviewer 2

I have read the manuscript by Sathe et al., entitled “Classical Criticality via Quantum Annealing”. They describe a transformative application of quantum annealing (QA) in classical statistical physics. The work is compelling and I recommend it to be published, if the following questions and comments can be addressed.

We thank the Reviewer for the appreciation of the significance of our work. Below, we address their comments.

Comment 1. The caption of Figure 1 states that the data “are in agreement with the exact solution.” Would it be more appropriate to phrase this as “qualitatively matches” instead? This adjustment would better reflect the subsequent quantitative analyses.

We appreciate the referee’s observation and suggestion. We have now change “agreement” to “qualitative agreement” in the caption of Figure 1.

Comment 2. Figure 2 references the fourth-order Binder cumulant U (Eq. 6) and its crossing points to extract J_c . To improve readability, consider relocating this validation to Section II (after introducing finite-size scaling formally).

We have now moved the figure so that it’s located closer to Eq. (6). The figure is now numbered 4, instead of 2.

Comment 3. The manuscript notes: “A slight scatter in intersection points is observed, consistent with finite-size deviations expected to arise from the limited system sizes imposed by hardware constraints.” Can you explain more about this? The trends in Fig. 2 suggest the line with larger system sizes do not necessarily converge toward J_c . Is this interpretation accurate?

We appreciate the referee’s comment. Since the system sizes probed in our analysis are relatively small, a sharp intersection point could not be obtained. We identified the crossing point as the location where most of the curves intersect. However, as the referee points out, the 12×12 system’s Binder cumulant curve gets close to, but does not cross the extracted crossing point for most s values. In contrast to the referee’s comment, we anticipate that the Binder cumulant intersections for larger system sizes will converge more closely to the true critical temperature. We believe that the deviation in the 12×12 system size curves is an effect primarily due to experimental errors and secondarily due to finite-size effects. Disentangling the relative contributions of experimental noise and deviations due to finite system sizes would be interesting and useful future work. However, this would require implementation of alternate temperature estimation techniques, which can be computationally expensive and were hence avoided in our study. Nonetheless, we have now added the following clarification in the updated manuscript:

(While the Binder cumulant curves should intersect at a single point in the thermodynamic limit, finite-size corrections can cause noticeable deviations [6, 2, 7]. We obtain a similar scatter in intersection points using MCMC on systems of similar sizes; see Methods.)

Here, we point the interested reader to two papers and a book that have documented finite-size corrections to the Binder cumulant, and other physical quantities close to critical points. In addition, we also show the Binder cumulant curves obtained using MCMC for similarly small system sizes, in Fig. 11a of the updated manuscript.

Comment 4. I notice that calibration refinement is essential for reliably measuring the quantities like magnetization susceptibility (Appendix A.3). This process reduce the effects of environmental noise—such as D-Wave’s intrinsic ~ 10 mK thermal noise—by adjusting flux-bias offsets (FBOs) and coupler strengths. Could this suppression artificially reduce thermal fluctuations critical to critical phenomena, potentially explaining why the QA-measured critical exponent ratio γ/ν deviates by $\sim 25\%$ from the theoretical value of 1.75 (Fig. 4d)?

We thank the referee for raising this important point. For our purposes, the noise and imperfections in the device manifests as uncorrelated local disorder, which has been partially validated by prior numerical studies [8], at least on small system sizes. The well-known Harris criterion states that if a disorder-free model has $\alpha < 0$, then the universality class does not change upon introducing uncorrelated local disorder, while for the $\alpha > 0$ case, the exponents can change. While the 2D Ising model (and the PUD model) lies at the marginal case of $\alpha = 0$, it is believed that the disorder effects do not change the critical exponents except for scaling corrections. However, these considerations can be pertinent for other models, and hence we state these explicitly in the updated manuscript:

Without calibration refinement, the device noise in the relevant parameter regimes manifests as disorder in both the longitudinal field and coupling strengths, with each anneal effectively sampling from a different

disorder realization [8]. The presence of such disorder can alter the critical exponents relative to those of the clean (i.e., disorder-free) model, particularly when the heat capacity exponent α for the clean system is greater than zero [9]. In the 2D Ising model (and the PUD model we consider), with $\alpha = 0$, weak uncorrelated disorder is believed to not change the critical exponents, though it may introduce logarithmic correction to scaling.

We thank the Reviewer for the careful reading and very useful suggestions that have helped improve the manuscript.

3 Reviewer 3

This manuscript presents the first demonstration of directly analyzing finite-temperature critical phenomena in a classical statistical model using quantum annealing (QA) hardware, representing a significant advance in the field. By tuning the Hamiltonian’s energy scale to control the effective temperature, the authors apply Binder cumulant analysis and finite-size scaling (FSS) to the Piled-Up Dominoes (PUD) model. This approach successfully reconstructs the phase diagram and estimates critical exponents, while also providing quantitative evidence that QA sampling exhibits little to no critical slowing down, a pronounced issue in classical Monte Carlo methods.

The work holds substantial significance for both statistical physics and quantum computing. While previous studies (e.g., Harris et al., *Science* 361, 162 (2018); King et al., *Nat. Commun.* 12, 1113 (2021)) have explored criticality with QA under specific conditions, this paper addresses a more challenging and tunably frustrated system, achieving hardware-based temperature control and phase diagram reconstruction. This sets a new benchmark for demonstrating the robustness and potential generality of the methodology. Moreover, the findings suggest that QA could circumvent computational bottlenecks encountered in classical methods near criticality, paving the way for broader applications.

We thank the Reviewer for the appreciation and for conceptualizing the significance of our work. Below, we address their comments.

1. The main claims, effective temperature control, phase diagram reconstruction, critical exponent extraction, and avoidance of critical slowing down, are generally supported by the presented data. However, the observed deviation of up to 25% in γ/ν from the theoretical value of 1.75 warrants further investigation to disentangle contributions from the energy scale–temperature approximation and finite-size effects.

We appreciate the referee’s insightful comment. The observed deviation of γ/ν (ranging between 15 to 25%) from the theoretical value is indeed non-negligible. However, such discrepancies are encountered when analyses are limited to relatively small system sizes, as in our present study. The finite-size scaling ansatz strictly holds only in the asymptotic regime of large L . Moreover, in our case, additional sources of uncertainty—such as the effective temperature calibration inherent to quantum annealing—can compound these finite-size effects. A precise separation of these contributions would require access to larger problem sizes, which we hope to pursue in future work as quantum hardware continues to improve.

In the second paragraph following Eq. (10) in the updated manuscript, we have now stated this explicitly:

This discrepancy may stem from deviations from the hypothesized relation, Eq. (9a), which was applied at two stages in the analysis, in addition to the relatively small system sizes used in our analysis. Understanding the relative contributions of these factors would require implementing alternative temperature estimation techniques, and access to a larger range of system sizes.

Demonstrating applicability to other models or types of phase transitions, such as Berezinskii–Kosterlitz–Thouless or first-order transitions, would further strengthen the generality of the conclusions, if the authors can.

We agree with the referee. However, studying entirely different models would lie outside the scope of the current manuscript. Hence, we state these as interesting future works in the Discussion section:

Investigating the performance of quantum annealers for studying other types of phase transitions, such as discontinuous ones or Berezinskii-Kosterlitz-Thouless phase transitions would be an interesting area for future research.

2. There are a few aspects of the data analysis and interpretation that could be improved. Notably, the error sources in critical exponent estimation have not been quantitatively separated, and no numerical before/after comparison of calibration refinement (shimming) is provided. While these issues do not constitute fatal flaws that preclude publication, clarifying them would enhance the paper’s credibility.

We appreciate the referee’s comment, and agree that a before/after comparison that demonstrates the impact of calibration refinement is important. In the updated manuscript, we have added such an analysis. In particular, we show the plots for the Binder cumulant, heat capacity and magnetization susceptibility when no calibration refinement is done, in Fig. 7 in the revised manuscript. For convenience, we also provide the accompanying discussion here:

The importance of the calibration refinement procedure is evident when comparing various physical quantities and observables, such as the Binder cumulant, heat capacity and magnetization susceptibility, obtained from quantum annealing without this refinement. In Fig. 7, we plot these quantities with the same annealing parameters as Fig. 2 in the main text, but without calibration refinement. In addition to the

prominent increase in the noise in the data, we find no peaks in the χ/β and C_V/β^2 plots, even after implementing experiments at very low energy scales ($J^{-1} \approx 25$). While peaks might appear at even lower energy scales, the associated measurements become increasingly noisy due to the finite precision with which such small energy differences can be implemented. Additionally, these peaks (if observed at all), would indicate a critical energy scale that departs significantly from values expected from the approximate transition region seen in the order parameter plot (Fig. 1c in the main text). In the same vein, we conclude that the crossing of the Binder cumulant curves seen in Fig. 7a, is inaccurate.

In addition, we have also included a before/after comparison of the magnetization and coupler frustration probability histograms in Fig. 8 of the revised manuscript.

We also now include a discussion regarding whether the noise in the device can significantly alter the critical exponents. (This point was also raised by another referee). In the main text, we have added the following paragraph:

Without calibration refinement, the device noise in the relevant parameter regimes manifests as disorder in both the longitudinal field and coupling strengths, with each anneal effectively sampling from a different disorder realization [8]. The presence of such disorder can alter the critical exponents relative to those of the clean (i.e., disorder-free) model, particularly when the heat capacity exponent α for the clean system is greater than zero [9]. In the 2D Ising model (and the PUD model we consider), with $\alpha = 0$, weak uncorrelated disorder is believed to not change the critical exponents, though it may introduce logarithmic correction to scaling.

3. The methodology is sound, with established statistical physics techniques such as Binder cumulant crossings and FSS applied appropriately. QA-specific procedures, including temperature tuning and embedding strategies, are reasonable and meet reproducibility standards in the field. The experimental conditions, sample sizes, system configurations, energy scale settings, and calibration steps (including fixed gauge transformation) are described in sufficient detail to enable replication. Providing more explicit numerical parameters or code snippets for the shimming procedure in the Supplementary Material would make replication even more straightforward.

We thank the referee for this suggestion. We have now included more details of the shimming procedure in the Methods section:

We implemented a smoothing of the FBOs and coupler values across the various energy scales starting at iteration number 75. The adaptive step size procedure was also implemented from iteration 75, with a look-back of 20 iterations to determine whether to increase or decrease the step sizes by a factor of 0.1. Shimming of coupler values was started at iteration 50, while the FBO shimming was started at step 20.

We believe that these details, in conjunction with “Procedure 1” that outlines the algorithm, are sufficient for the reader to reproduce our results. Additionally, our procedure is primarily a modification of ones presented in Ref. [10] (also cited in the manuscript) and the accompanying GitHub repository.

We have also now added plots (Fig. 8 in the revised manuscript) that illustrate the success of the calibration refinement protocol in achieving lower spreads in the magnetization and coupler frustration probabilities.

Overall, this manuscript exhibits high originality and scholarly value. With the suggested clarifications and minor enhancements, it would be a strong candidate for publication in Nature Communications.

We thank the Reviewer for their careful reading and appreciate the very useful suggestions that have helped improve the manuscript.

References

- [1] J. Villain, R. Bidaux, J.-P. Carton, and R. Conte. Order as an effect of disorder. *Journal de Physique*, 41(11):1263–1272, 1980.
- [2] Kurt Binder and Dieter W. Heermann. *Monte Carlo Simulation in Statistical Physics: An Introduction*, volume 0 of *Graduate Texts in Physics*. Springer, Berlin, Heidelberg, 2010. <https://link.springer.com/10.1007/978-3-642-03163-2>.
- [3] Roderich Moessner and Arthur P. Ramirez. Geometrical frustration. *Physics Today*, 59(2):24–29, February 2006.

- [4] Hilal Saglam, Ayhan Duzgun, Aikaterini Kargioti, Nikhil Harle, Xiaoyu Zhang, Nicholas S. Bingham, Yuyang Lao, Ian Gilbert, Joseph Sklenar, Justin D. Watts, Justin Ramberger, Daniel Bromley, Rajesh V. Chopdekar, Liam O'Brien, Chris Leighton, Cristiano Nisoli, and Peter Schiffer. Entropy-driven order in an array of nanomagnets. *Nature Physics*, 18(6):706–712, June 2022.
- [5] Forthcoming work.
- [6] K. Binder. Finite size scaling analysis of Ising Model block distribution functions. *Zeitschrift für Physik B Condensed Matter*, 43(2):119–140, June 1981. <https://doi.org/10.1007/BF01293604>.
- [7] Alan M. Ferrenberg and D. P. Landau. Critical behavior of the three-dimensional Ising model: A high-resolution Monte Carlo study. *Physical Review B*, 44(10):5081–5091, September 1991.
- [8] Marc Vuffray, Carleton Coffrin, Yaroslav A. Kharkov, and Andrey Y. Lokhov. Programmable Quantum Annealers as Noisy Gibbs Samplers. *PRX Quantum*, 3(2):020317, April 2022. <https://link.aps.org/doi/10.1103/PRXQuantum.3.020317>.
- [9] A. B. Harris. Effect of random defects on the critical behaviour of Ising models. *Journal of Physics C: Solid State Physics*, 7(9):1671, May 1974.
- [10] Kevin Chern, Kelly Boothby, Jack Raymond, Pau Farré, and Andrew D. King. Tutorial: Calibration refinement in quantum annealing. *Frontiers in Computer Science*, 5, 2023. <https://www.frontiersin.org/articles/10.3389/fcomp.2023.1238988>.